# Characterization of the *Neurospora crassa* Galactosaminogalactan Biosynthetic Pathway

**DOI:** 10.3390/microorganisms12081509

**Published:** 2024-07-23

**Authors:** Apurva Chatrath, Protyusha Dey, Kevin Greeley, Gabriela Maciel, Lei Huang, Christian Heiss, Ian Black, Parastoo Azadi, Stephen J. Free

**Affiliations:** 1Department of Biological Sciences, SUNY University at Buffalo, Buffalo, NY 14260, USA; apurva.chatrath@gmail.com (A.C.); kagreele@buffalo.edu (K.G.);; 2Complex Carbohydrate Research Center, University of Georgia, Athens, GA 30602, USA; lei28@uga.edu (L.H.);

**Keywords:** galactosaminogalactan, adhesin, polysaccharide synthase, polysaccharide deacetylase, *Neurospora crassa*, amino sugar determination, polysaccharide composition

## Abstract

The *Neurospora crassa* genome has a gene cluster for the synthesis of galactosaminogalactan (GAG). The gene cluster includes the following: (1) UDP-glucose-4-epimerase to convert UDP-glucose and UDP-*N*-acetylglucosamine to UDP-galactose and UDP-*N*-acetylgalactosamine (NCU05133), (2) GAG synthase for the synthesis of an acetylated GAG (NCU05132), (3) GAG deacetylase (/NCW-1/NCU05137), (4) GH135-1, a GAG hydrolase with specificity for *N*-acetylgalactosamine-containing GAG (NCU05135), and (5) GH114-1, a galactosaminidase with specificity for galactosamine-containing GAG (NCU05136). The deacetylase was previously shown to be a major cell wall glycoprotein and given the name of NCW-1 (non-GPI anchored cell wall protein-1). Characterization of the polysaccharides found in the growth medium from the wild type and the GAG synthase mutant demonstrates that there is a major reduction in the levels of polysaccharides containing galactosamine and *N*-acetylgalactosamine in the mutant growth medium, providing evidence that the synthase is responsible for the production of a GAG. The analysis also indicates that there are other galactose-containing polysaccharides produced by the fungus. Phenotypic characterization of wild-type and mutant isolates showed that deacetylated GAG from the wild type can function as an adhesin to a glass surface and provides the fungal mat with tensile strength, demonstrating that the deacetylated GAG functions as an intercellular adhesive. The acetylated GAG produced by the deacetylase mutant was found to function as an adhesive for chitin, alumina, celite (diatomaceous earth), activated charcoal, and wheat leaf particulates.

## 1. Introduction

The fungal cell wall is a dynamic organelle which protects the cell from environment stress and allows the cell to interact with the environment. The cell wall is largely composed of glucans and glycoproteins. A variety of glucans have been observed in cell walls. Chitins, β-1,3-glucans, mixed β-1,3/1,4-glucans, and α-1,3-glucans have been shown to be cross-linked together to create a cell wall polysaccharide matrix. In filamentous fungi, galactosaminogalactans (GAGs) have also been shown to be cell wall elements. GAGs are long polymers of α-1,4-linked galactose, galactosamine, and *N*-acetylgalactosamine residues which are thought to adopt an elongated, nearly straight confirmation with the amino groups (*N*-acetylgalactosamine and galactosamine) pointing to the solvent and available for binding partners [1,2]. In *N. crassa*, GAGs were observed in the growth medium by Reissig and Glasgow [3], and a GAG deacetylase was briefly studied by Jorge et al. [4]. The enzymes for the GAG biosynthetic pathway are encoded in a five-gene cluster in a variety of ascomycete genomes [5]. These enzymes include a GAG synthase, a UDP-glucose 4 epimerase (to convert UDP-glucose and UDP-*N*-acetylglucosamine to UDP-galactose and UDP *N*-acetylgalactosamine), a GH135 family GAG hydrolase (a spherulin 4-like hydrolase called sph3), a GH114 family galactosaminidase, and an *N*-acetylgalactosamine deacetylase.

The literature shows that the GAG from the Aspergilli is a heteropolymer containing galactose, galactosamine, and *N*-acetylgalactosamine residues [2]. The *Aspergillus fumigatus* GAG is initially synthesized with acetylated galactosamines, and the deacetylase then partially deacetylases the GAG to create a polycationic polymer. In *A. fumigatus*, GAG is essential for various pathogenic processes such as virulence, biofilm formation, antibiotic resistance, immune modulation and evasion, adhesion to host cells, adhesion to plastics, and adhesion to negatively charged surfaces [5,6,7,8,9,10,11,12]. GAG has been shown to mask the β-1,3-glucan elements in the cell wall to shield them from the host immune system [9,13]. GAGs have also been shown to be required for cell-to-cell adhesions and aggregation of *Aspergillus oryzae* [14,15]. In the insect pathogenic fungus *Metarhizium robertsii*, a GAG has also been found to function as an adhesive to the insect cuticle and to be needed for biofilm formation [16].

Since GAG biosynthetic clusters are common in the ascomycetes [5,6], including many fungi that are nonpathogenic fungi (saprophytes) and plant pathogens, the question of what advantages GAGs offer to these fungi remains to be determined. In our studies on the *N. crassa* cell wall, we noticed that one of our major cell wall proteins, NCW-1 (NCU05137) [17,18,19], is a homolog of the *A, fumigatus* GAG deacetylase Agd-1. The *A. fumigatus* deacetylated GAG was shown to be required for biofilm formation and for the hyphae to adhere to each other [5,7,14,15]. To better characterize the *N. crassa* cell wall and ask what advantages a GAG might offer to saprophytes and plant pathogens, we characterized the *N. crassa* GAG biosynthetic pathway. Characterization of the polysaccharides from the growth medium provided evidence of the GAG synthase functioning to produce a polysaccharide containing galactosamine and *N*-acetylgalactosamine. We found that the loss of the GAG synthase affected the tensile strength of the fungal mycelia and its ability to adhere to a glass surface, alumina, cellite, activated charcoal, and wheat grass. The *N. crassa* GAG hydrolase (NCU05136) was found to have hydrolytic activity towards a polysaccharide found in the growth medium. We found evidence suggesting that both the acetylated and deacetylated forms of GAGs are polysaccharide adhesins. In our studies, we found that the deacetylated GAG produced by wild-type hyphae mediates adhesion to glass surfaces and provides the fungal network tensile strength, suggesting that the deacetylated GAG facilitates intercellular interactions within the hyphal network. We found that the acetylated GAG produced by the deacetylase mutant mediated adhesion to chitin, alumina, celite (diatomaceous earth), wheat grass, and activated charcoal particulates. We believe that the acetylated GAG and chitin are both synthesized near the edge of the growing colony and the ability of the acetylated GAG to adhere to chitin may facilitate adherence of the GAG to the cell wall. We hypothesize that the ability of the acetylated GAG to adhere to wheat grass particulates and similar plant-derived material could facilitate the digestion of such materials by saprophytes and plant pathogens. The ability of the acetylated GAG to bind to activated charcoal (burnt plant material) is especially interesting for *N. crassa*, which grows prolifically on burnt trees following forest fires. We propose that the acetylated GAG may facilitate the adhesion of the fungus to natural substrates such as burnt wood and leaf debris present in the soil.

## 2. Materials and Methods

### 2.1. Strains and Growth Conditions

*N. crassa* strains were obtained from the Fungal Genetics Stock Center. The mutant strains used were from the Neurospora deletion library [20]. A listing of the strains is provided in Table 1. The strains were routinely cultured on Vogel’s sucrose minimal medium [21]. Homokarytic deletion mutant isolates for NCU05132, NCU05135, and NCU05137 were available from the deletion library. To verify that the deletion mutants from the library contained the specified deletions, primers for amplifying the 5′ end of the normal and deletion mutant genes were generated for NCU05132, NCU5136, and NCU5137 (Table 2). Small-scale DNA preparations from the deletion mutants were prepared and used for PCR reactions. PCR reactions with the gene-specific forward primer (labelled XF) and a hygromycin-specific reverse primer (HygroR) were used to amplify the deletion mutant gene and PCR reactions with the gene-specific forward and reverse primers (labelled XF and XR) were used to amplify the normal gene (Table 2). Analysis of the ∆NCU05132 and ∆NCU05137 demonstrated that these deletion mutants contained the correct deletions (Appendix A). The NCU05136 deletion mutant was found as a heterokaryon (isolate with both wild-type and deletion mutant nuclei) in the deletion library. PCR analysis of the genomic DNA from hygromycin-resistant progeny from the heterokaryon showed that the NCU05136 gene was not deleted, and the mutant was not further characterized.

### 2.2. Cloning HIS6-Tagged NCU05136

An HIS6-tagged version of the GAG hydrolase (NCU05136) coding region was generated, purified, and used to examine its enzymatic activities. Primers 5136F and 5136H6R were used to PCR-amplify the GAG hydrolase coding region with the HIS6 tag located at the carboxyl terminus (Table 2). The PCR product was purified by electrophoresis in an agarose gel, isolated using a QIAquick gel extraction kit (Qiagen, Hilden, Germany), and cloned into the pJET vector (Thermo Scientific, Carlsbad, CA, USA). The cloned genes were then subcloned into a pMF272 vector [22] that had been digested with BamHI and EcoRI to create the p5136H6 plasmid. The plasmid contained a portion of the *his-3* coding region, the *ccg-1/grg-1* promoter, the cloned coding regions for NCU05136, and the 3′ utr from the *his-3* gene. The *his-3* coding region and 3′utr allowed for insertion of plasmid DNA sequences into the *N. crassa* genome in the 3′utr region of the *his-3* gene by homologous recombination [22]. Insertion of the plasmid into the *his-3* locus of an *his-3* mutant generated a histidine auxotroph and allowed for the easy isolation of transformants. The *ccg-1/grg-1* promoter allowed for high-level expression of the tagged GAG hydrolase [23]. The p5136H6 plasmid was used in transformation experiments to generate *N. crassa* strains expressing the H6-tagged GAG hydrolase.

### 2.3. Purification of the H6-Tagged GAG Hydrolase (NCU05136)

Transformant strains expressing the GAG hydrolase were grown at 30 °C for 48 h at room temperature in a liquid Vogel’s 2% sucrose medium in a shaker incubator (120 rpm). The cells were harvested on a Buchner funnel, ground to a fine powder under liquid nitrogen with a mortar and pestle, and suspended in 5 mL 10 mM imidazole, 50 mM NaPO4, and 300 mm NaCl. Following a centrifugation step (12,000× *g* for 10 min), the supernatant was loaded onto a 1 mL nickel Probond column (ThermoFischer, Waltham, MA, USA), and the column was washed with 25 mL of the suspension buffer. The column was then washed with 10 mL of 25 mM imidazole, 50 mM NaPO_4_, and 300 mm NaCl. The GAG hydrolase was eluted with a liner gradient of imidazole (25 mM to 250 mM) in 50 mM NaPO_4_, and 300 mm NaCl, and the factions containing the purified protein were identified by Coomassie stained PAGE and by Western blot analysis. The purification steps on the Probond column were carried out at 4 °C.

### 2.4. Assays Testing Adhesive Properties of Hyphal Networks

To assess whether a particulate could adhere to hyphae, wild-type and mutant isolates were inoculated at the edge of a Petri dish containing Vogel’s sucrose agar medium and allowed to grow at 30 °C for between 16 and 20 h. Alumina, activated charcoal, and celite particulates were suspended in distilled water at a concentration of 10 mg/mL, and 10 mls was added to the Petri dish and allowed to sit for 30 min. The plate was then gently swirled, and the liquid, with any loose particulates, was poured off. The plates were then washed seven times by gently adding 10 mL of water, gently swirling, and dumping the water with any loose particulates. The plates were then allowed to dry for about 30 min and photographed. Adhesion of wheat grass to the hyphae was assessed in the same manner, except that the initial concentration of wheat grass was 3 mg/mL. Alumina was obtained from Sorbtech (Norcross, GA, USA). Celite and activated charcoal were obtained from Sigma-Aldrich (St. Louis, MO, USA). Wheat grass was obtained from Pines International, Inc. (Lawrence, KS, USA).

To determine if the presence of the GAG affected the tensile strength of a hyphal mat, wild-type and mutant strains were grown at 30 °C for 5 days in 25 mL of liquid Vogel’s sucrose in a 125 Erlenmeyer flask with an aluminum foil cap. The hyphal mats, which formed on the surface of the medium, were collected on a Buchner funnel and blotted dry. The hyphal mats were folded twice to give a thickness four times that of the original mat and placed in a penetrometer apparatus (Appendix A), and the tensile strength of the hyphal mat was determined. The penetrometer (model FT 02) was purchased from QA Supplies (Norfolk, VA, USA). The home-made apparatus was constructed from 2 sheets of plexiglass with an array of nine 3 mm holes drilled into the plexiglass sheets. A 2 mm probe was used in the penetrometer for measuring hyphal tensile strength.

A centrifugation assay was used to examine the ability of hyphae to bind to a glass surface. Wild-type and mutant hyphae were grown in 5 mL of Vogel’s sucrose medium in a 30 mL Corex centrifuge tube for 3 days at 30 °C in a slant rack and then the Corex tube was placed in an upright position for another 24 h to allow hyphae the grow up the sides of the Corex tube. The tubes were then photographed for a “pre-centrifugation” image. The tubes were carefully filled with water to near the top and centrifuged at 2500× *g* for 5 min in a swinging bucket rotor. After centrifugation, the water was carefully aspirated from the centrifuge tubes and the tubes were photographed for a “post-centrifugation” image. The ability of the hyphae to adhere to the glass Corex tube was determined by examining the sides of the centrifuge tubes for the presence of hyphae.

### 2.5. Assays for GAG Hydrolase Activity by TLC (Thin-Layer Chromatography) and PACE (Polysaccharide Analysis Using Carbohydrate Gel Electrophoresis)

Substrates for a GAG hydrolase assay are not commercially available, so we used polysaccharides collected from the growth media in which wild-type and mutant cells had grown as substrates for our assays. To test for GH114-1 hydrolase activity, we incubated our purified H6-tagged GH114-1 hydrolase (NCU05136) with polysaccharides from the wild type. The polysaccharides were obtained by growing the strains in 400 mL of liquid Vogel’s 2% sucrose medium at 30 °C for 3 days in a shaker incubator and collecting the growth medium in a Buchner funnel. Ethanol was added to a final concentration of 70%, and the polysaccharides were allowed to precipitate at 4 °C for 2 days. The precipitated polysaccharides were collected by centrifugation (8000× *g* for 10 min) and washed with 70% ethanol. The precipitated polysaccharides were then resuspended in 800 µL water. The assay for GAG hydrolase included 5 µL of purified GH114-1, 22 µL of polysaccharide and 3 µL of 1 M sodium acetate pH 5.0 buffer. The hydrolase and polysaccharides were incubated at room temperature for 24 h, and the hydrolysis of GAG was assessed by thin-layer chromatography (TLC) and PACE gel electrophoresis. For TLC analysis, the sample was spotted onto a TLC plate (TLC silica gel 60 from Merck, Darmstadt, Germany), and chromatography was performed with a (5:2:3) mixture of N-butanol:acetic acid:water solvent for 3.5 h. The TLC plates were then stained by spraying with orcinol reagent (1% orcinol in 70:3 mixture of ethanol:sulfuric acid) and heating to 100 °C for 15 min. For PACE analysis, the digested polysaccharides were labelled at their reducing ends with ANTS (8-aminonaphthalene-1,3,6-trisulfonic acid) and subjected to electrophoresis in a 15% polyacrylamide gel as described by Pidatala et al. [24].

### 2.6. Compositional Analysis of Dried Polysaccharides from the Growth Medium via Methanolysis and TMS Derivatization

To determine if polysaccharides from the growth medium contained galactose and *N*-acetylgalactosamine, a carbohydrate composition determination was carried out on the polysaccharides released by wild-type, ∆NCU05132 (GAG synthase) and ∆NCU05137 (GAG deacetylase) cells grown in 100 mL of Vogel’s sucrose liquid medium in a shaker incubator (120 rpm) at 30 °C for three days. Polysaccharides were collected from the growth medium with a 70% ethanol precipitation, washed with a 70% ethanol wash, air-dried, and subjected to a carbohydrate compositional analysis. Glycosyl composition analysis was performed by combined gas chromatography–mass spectrometry (GC-MS) of the per-O-trimethylsilyl (TMS) derivatives of the monosaccharide methyl glycosides produced from the sample by acidic methanolysis as described previously by Coleman et al. [25]. Briefly, 200–400 mg dried samples were heated with 1 M methanolic HCl in a sealed screw-top glass test tube for 18 h at 80 °C. After cooling and removal of the solvent under a stream of nitrogen, the samples were re-N-acetylated by treatment with acetic anhydride and pyridine in methanol and dried again. The samples were then derivatized with Tri-Sil^®^ (Thermo) at 80 °C for 30 min. GC-MS analysis of the TMS methyl glycosides was performed on an Agilent 7890A GC interfaced to a 5975C MSD, using a Supelco Equity-1 fused silica capillary column (30 m, 0.25 mm ID) with a temperature gradient.

### 2.7. Compositional Analysis of Concentrated Polysaccharide via Nitrous Acid Digestion and Generation of Alditol Acetates

Because a large amount of medium polysaccharide was not soluble following ethanol precipitation or drying, we also obtained medium polysaccharide samples via an evaporation protocol. Growth media from the wild type, GAG synthase mutant, and GAG deacetylase mutant were collected on a Buchner funnel, and 100 mL of medium was evaporated down to a volume of 5 mL in a speed vac. The samples were then dialyzed against water and further concentrated to 2 mL in the speed vac. An analysis of the galactosamine and N-acetylated galactosamine within the polysaccharides was determined by employing nitrous acid conversion of the galactosamine to 2,5 anhydrotalose and then generation of alditol acetates from all of the sugar residues in the samples [26]. Briefly, 100 µL of the concentrated polysaccharide samples were combined with 100 µL 0.3 M nitrous acid solution (492 µL nonpure water, 16 µL acetic acid, 269 mg sodium nitrite). The solutions were allowed to react at room temperature for 50 min. Next, 250 µL of a sodium carbonate solution (100 mg/mL in water) was added to neutralize, and the samples were reduced by addition of 250 µL sodium borohydride solution (60 mg/mL in water). Reduction was carried out overnight at room temperature. The samples were then neutralized by addition of 100 µL acetic acid and dried under nitrogen to remove the nitrous acid and acetic acid. Finally, the samples were lyophilized to dryness. To separate the carbohydrate from contaminating salt, the samples were acetylated by addition of 500 µL each of pyridine and acetic anhydride to the dried samples. The samples were heated to 100 °C for 1 h. The acetylated carbohydrates were then extracted using a mixture of equal parts water and dichloromethane. The dichloromethane layer was washed 5 times with fresh water to remove all the salt. Finally, the dichloromethane layer containing the acetylated sample was transferred to a fresh tube and dried. Alditol acetates were then prepared from the samples via hydrolysis in trifluoroacetic acid (400 µL, 2 M TFA, 120 °C, 2 h), reduction with sodium borodeuteride (300 µL of a 10 mg/mL solution in water, 4 h at room temperature), and re-acetylation (250 µL of both pyridine and acetic anhydride, 100 °C, 1 h). The derivatized samples were again extracted using dichloromethane, and the resultant alditol acetates were run on an Agilent 7890A GC interfaced to a 5975C MSD (mass selective detector, electron impact ionization mode); separation was performed on a Supelco Equity-1 fused silica capillary column (30 m × 0.25 mm ID). Alongside the samples, a standard containing glucose, galactose, mannose, galactosamine, and N-acetyl galactosamine was run. To the samples and standard, ribose was added as an internal standard.

## 3. Results

### 3.1. N. crassa Contains a Galactosaminogalactan Gene Cluster

The *N. crassa* genome contains a gene cluster involved in the biosynthesis of cell wall galactosaminogalactan, similar to the galactosaminogalactan gene clusters found in a variety of other fungi [5]. As shown in Figure 1, the 30 kb gene cluster contains six genes: (1) NCU05132, a galactosaminogalactan synthase, (2) NCU05133, a UDP-glucose 4-epimerase that can convert UDP-glucose to UDP-galactose and N-acetylglucosamine to *N*-acetylgalactosamine [27], (3) NCU05134, a small gene of unknown function having a signal peptide, (4) NCU05135, a GH family 135 glycosyl hydrolase with specificity for *N*-acetylgalactosamine polymers [28,29], (5) NCU05136, an endo-α-1,4-galactosamidase [30], and (6) NCU05137, an *N*-acetylgalactosamine deacetylase [5,7]. The small gene of unknown function is found only in a few closely related fungal genomes while the other five genes are found encoded in GAG gene clusters in other filamentous fungi. The galactosaminogalactan synthase functions to generate long linear α-1,4-linked polysaccharide with *N*-acetylgalactosamine and galactose residues [2,9,12]. GH135-1 is a homolog of sph3, a spherulin-4 protein, which has been shown to be a GAG hydrolase with a TM domain near its N terminus. In *A. fumigatus*, sph3 has been found to be required for the synthesis of the GAG polysaccharide [28]. Sph3 has been shown to be capable of digesting biofilms and has specificity for *N*-acetylgalactosamine-containing polymers [29]. GH114-1 is an endo-α-1,4-galactosaminidase and has been characterized and shown to have specificity for galactosamine-containing (deacetylated) polysaccharide [30]. The *A. fumigatus* deacetylase has been shown to partially deacetylate the polysaccharide to generate a cationic polysaccharide able to act as an adhesin and needed for biofilm formation [5,7]. In our characterizations of the *N. crassa* cell wall, we found the deacetylase to be a major cell wall glycoprotein, and we named the gene *ncw-1* [17,18,19]. Deletion mutants for the GAG synthase and GAG deacetylase genes were obtained from the Neurospora deletion library and used for an analysis of the GAG biosynthetic pathway. A PCR analysis of the genomic DNA from these mutants demonstrated that the mutants contained the correct gene deletions and could be used to analyze mutant phenotypes (Appendix A).

### 3.2. Carbohydrate Composition Analyses of Polysaccharides

To characterize the *N. crassa* GAG, we isolated polysaccharides from the growth medium for the wild type, ∆NCU05132 (GAG synthase), and ∆NCU05137 (GAG deacetylase) by ethanol precipitation and resuspended the polysaccharides in water. We found that a fraction of the polysaccharides remained insoluble after the precipitation step. We subjected the soluble polysaccharides to a carbohydrate compositional analysis. The results of this analysis are found in Table 3. The compositional analysis shows that the soluble polysaccharides from the wild-type isolate, ∆NCU05132, and ∆NCU05137 contained mannose, glucose, and galactose (Table 3). The major difference between the polysaccharide compositions is that the soluble polysaccharides from the deacetylase mutant contain *N*-acetylgalactosamine and/or galactosamine residues (the analysis is unable to discriminate between *N*-acetylgalactosamine and galactosamine). The soluble polysaccharides from the wild type and ∆NCU05132 mutant were totally devoid of *N*-acetylgalactosamine. The lack of *N*-acetylgalactosamine and galactosamine in the GAG synthase mutant was expected, since the mutant should be unable to produce GAG. The lack of *N*-acetylgalactosamine and galactosamine in the soluble polysaccharides from the wild type was unexpected. The analysis suggests that the GAG from the wild-type cell may remain insoluble while some of the acetylated GAG from the deacetylase mutant was soluble. The presence of galactose in the soluble polysaccharides from the GAG synthase deletion mutant indicates that, in addition to synthesizing GAG, *N. crassa* produces at least one additional galactose-containing polysaccharide and that some of the polysaccharide is found in the soluble polysaccharide fraction following precipitation with 70% ethanol. With the polysaccharides forming a partition between the insoluble and soluble fractions, this analysis was unable to provide a complete picture of the polysaccharides found in the growth medium. However, the data do demonstrate there are differences in the polysaccharides produced by the wild type and the mutants and suggests that the deacetylase might function to deacetylate an *N*-acetylgalactosamine-containing polysaccharide found in the growth medium.

Since a major portion of the medium polysaccharides were insoluble following the ethanol precipitation, we attempted to obtain a compositional analysis of the medium polysaccharides by concentrating the polysaccharides with a speed vac and then carrying out an analysis that discriminated between galactosamine and *N*-acetylgalactosamine. The results of the analysis are shown in Table 4. As expected, all the samples contained mannose and glucose. We found that there was a major reduction in the amounts of galactosamine and *N*-acetylgalactosamine in the polysaccharides from the GAG synthase mutant, providing evidence that the synthase produces GAG. However, the polysaccharides from the GAG synthase mutant contained large amounts of galactose as well as very small amounts of galactosamine and *N*-acetylgalactosamine, suggesting that *N. crassa* synthesizes other polysaccharides containing these sugars. The polysaccharides from the wild type and GAG deacetylase mutant contain polysaccharides containing galactose, galactosamine, and *N*-acetylgalactosamine. Surprisingly, the amount of *N*-acetylgalactosamine was higher in the wild type than in the deacetylase mutant. With the fungus producing multiple polysaccharides containing galactose, galactosamine, and *N*-acetylgalactosamine, we were unable to demonstrate that the deacetylase mutant specifically deacetylates GAG with this analysis. The *N. crassa* genome contains several uncharacterized polysaccharide synthases that might be responsible for the galactose-containing polysaccharides we observed in the analysis.

The composition of the polysaccharides from growth medium in which wild-type cells, ∆NCU05132 cells, and ∆NCU05137 cells is shown in Table 4. The cells were grown in liquid shaking culture, and the medium was collected, concentrated, and analyzed. Analysis of free versus N-acetylated galactosamine was determined by employing nitrous acid conversion of galactosamine to 2,5-anhydrotalose. The mole percentage for each of the observed sugars is shown.

### 3.3. Demonstration of Endo-α-1,4-galactosaminidase (GAG Hydrolase) Enzymatic Activity

To look at the function of the endo-α-1,4-endogalactosaminidase, GH114-1, we cloned an HIS-6 version of NCU05136, expressed the tagged enzyme in *N. crassa*, and purified the protein. Figure 2A shows an image of the purified GH114-1 in a Coomassie-stained polyacrylamide gel. We used polysaccharides obtained from the growth medium from the wild type as a substrate for a GAG hydrolase assay. The polysaccharides were incubated with the purified enzyme, and the production of smaller oligosaccharides was assessed on a TLC plate and with PACE. Figure 2B is an image from a TLC analysis which shows that a smaller oligosaccharide was produced when the polysaccharides from the growth medium were subjected to digestion by the purified enzyme. Figure 2C is an image from a PACE analysis showing that digestion of the polysaccharide sample with the purified GH114-1 generated smaller oligosaccharides. The TLC and PACE analyses included maltodextrins as positive control and to give a rough estimation of oligosaccharide sizes. Our results suggest that the most abundant oligosaccharide generated by GH114-1 digestion of the polysaccharides from the wild-type growth medium is likely to contain four or five sugar residues. Experiments in which the purified enzyme was incubated with the chromogenic substrate p-nitrophenyl-α-D-galactopyranoside showed that the enzyme could not hydrolyze the chromogenic substrate, consistent with the finding that the enzyme is an endo glycosyl hydrolase. Lacking a purified substrate, we were unable to identify or further characterize the released oligosaccharide, but the results demonstrate that NCU05136 has polysaccharide hydrolase activity.

### 3.4. Phenotypic Characterization of GAG Mutants

As a measure of the ability of the hyphae to adhere to each other within a hyphal network, we tested the tensile strength of wild-type and mutant hyphal mats using a penetrometer (Appendix A). We found that it required a force of 290 ± 45 g to penetrate wild-type hyphal mats (mean ± SD from 7 independent determinations). It required a force of 212 ± 56 g to penetrate hyphal mats from the GAG synthase mutant (∆NCU05132) and 186 ± 49 g to penetrate hyphal mats from the deacetylase mutant (∆NCU05137) (means ± SDs from 8 independent determinations). This represents a 27% reduction in tensile strength for hyphal mats from the GAG synthase mutant and a 36% reduction in tensile strength for hyphal mats from the deacetylase mutant. Four different series of these penetrometer assays were carried out with similar results, demonstrating that the results were reproducible. We conclude that the deacetylated GAG facilitates the formation of a stronger hyphal network.

To look at the ability of hyphae to adhere to a glass surface, we devised a centrifuge test in which hyphae were grown in a 30 mL Corex centrifuge tube, and the ability of the hyphae to remain adherent to the sides of the tube when subjected to a centrifugal force was assessed. Wild-type, ∆NCU05132 (∆GAG synthase), ∆NCU05135 (∆GH135 hydrolase), and ∆NCU05137 (∆GAG deacetylase) hyphae were grown in the centrifuge tubes and subjected to centrifugation at 2500× *g*. As shown in Figure 3, wild-type hyphae and the hyphae from ∆NCU05135 remained adherent while the ∆NCU05132 and ∆NCU05137 hyphae were unable to remain adherent during the centrifugation. This centrifugation assay to assess the ability of the hyphae to retain adherence to the centrifuge tube was repeated six times to demonstrate that the results shown in Figure 3 were reproducible. We conclude that the deacetylated GAG found in the wild type and NCU05135 mutant function to mediate adhesion to a glass surface. The GAG synthase mutant, lacking GAG, was unable to adhere to the glass, and the acetylated GAG in the deacetylase mutant did not function as an adhesin. Our results showing that the deacetylated GAG functions as an adhesin are similar to those of others showing that GAG from Aspergilli mediate cell-to-cell interactions, adhesion to host, and biofilm formation [5,7,14,15].

To examine the ability of hyphae to adhere to a variety of particulates, we developed a particulate binding assay in which particulates in water were added to an agar plate with an actively growing hyphal colony and, after a short period of time in which the particulates could bind to the hyphae, the unbound particulates were washed away. We found that there were dramatic differences between hyphae lacking GAG, hyphae with acetylated GAG, and hyphae with deacetylated GAG in their ability to bind particulates. In Figure 4, we show the binding of wheat grass, alumina, celite (diatomaceous earth), activated charcoal, and chitin particulates to the three types of hyphae. We found that hyphae with the acetylated GAG from the ∆NCU05137 (GAG deacetylase mutant) effectively bound to all five of these particulates. Wild-type hyphae with the deacetylated GAG and hyphae from ∆NCU05132 (GAG synthase mutant) lacking GAG bound significantly less of the particulate than did the ∆NCU05137 mutant. The small amounts of particulate bound by the synthase mutant and wild-type hyphae was found near the growing edge of the colony. Higher-magnification images show that the particulates are binding to the hyphae (Appendix A). Five independent particulate binding experiments were carried out to demonstrate that the binding of these particulates was reproducible. Other particulates tested for this assay included silica, glass powder, bentonite, xylan, and cellulose, but these did not show significant binding to any of the hyphae. The five particulates that did bind to the fully acetylated GAG are representative of things the fungus might encounter in its natural environment. We believe that GAG and chitin are both synthesized and secreted at the hyphal tips near the growing edge of the hyphal colony and the GAG binding to chitin could anchor the GAG into the cell wall and allow GAG to remain associated with the wall. The binding to activated charcoal is particularly interesting, since *N. crassa* is renowned for its ability to colonize burnt trees after a forest fire. Wheat grass is representative of plant materials *N. crassa* might encounter as a saprophyte. Adherence to plant-derived materials could facilitate the digestion of the materials and the uptake of the released sugars and amino acids by having them released close to the plasma membrane. Our results with the particulate binding assay add to the characterization of GAGs and their functions. While our Corex tube binding assay, our hyphal mat tensile strength assay, and the results from other studies show that the deacetylated GAG is an adhesin, the particulate binding assays show that the acetylated form of GAG can also function as an adhesin. The results show that two different forms of the GAG have different binding specificities that allow them to adhere to different materials.

## 4. Discussion

GAGs are long linear α-1,4-linked polysaccharides containing galactose, galactosamine, and *N*-acetylgalactosamine residues [1,2]. They have been shown to play important roles in cell-to-cell interactions, the formation of biofilms, and in human and insect pathogenesis [5,6,7,8,9,11,12,16]. A cluster of five genes involved in the synthesis of GAGs has been shown to be present in a large variety of ascomycetes (Figure 1) [5]. The five gene cluster contains the following: (1) a UDP-glucose 4-epimase to convert UDP-glucose and UDP-*N*-acetylglucosamine to UDP-galactose and UDP-*N*-acetylgalactosamine, which are needed for the synthesis of the polysaccharide, (2) a GAG synthase which generates an acetylated GAG, (3) a GAG deacetylase which partially deacetylates the GAG to produce a polycationic polysaccharide, (4) a GH135 GAG hydrolase (sph3 spherulin 4-like protein) which is needed for GAG synthesis and is able to hydrolyze acetylated GAGs, and (5) a GH114 endo-α-1,4-galactosaminidase able to cleave deacetylated GAG. While GAG synthesis and function have been extensively studied in the human pathogen *A. fumigatus*, the role of the GAG pathway has not been carefully examined in saprophytic fungi and fungal plant pathogens. While *N. crassa* is generally considered to be a saprophyte, it is closely related to several plant pathogens and has been shown to be a pathogen in conifers [31]. We have examined the role of the pathway in *N. crassa* by characterizing the polysaccharides produced by mutants affected in the pathway and by characterizing the phenotypic consequences of losing pathway enzymes.

GAG was first identified in *N. crassa* by Reissig and Glasgow [3] and the GAG deacetylase has been partially purified and characterized [4]. However, the composition and structure of the GAGs were not elucidated in these early studies. Work in *A. fumigatus* demonstrated that GAG is a heteropolymer containing galactose, *N*-acetylgalactosamine and galactosamine and that the deacetylase was needed for the synthesis of what was thought to be the functional product of the pathway, a partially deacetylated GAG [2,6,7].

The activity of EGA3, the *A. fumigatus* homolog of *N. crassa* GH114-1 endo-α-1,4-galactosaminidase, has been previously examined [30]. Bamford et al. determined the crystal structure of the protein, identified the active site and a number of important interactions involved in its enzymatic activity, and demonstrated that it was an endo-α-1,4-galactosaminidase able to recognize deacetylated GAG [30]. The *N. crassa* enzyme is encoded by NCU05136 and, like the enzymes encoded in other fungal genomes, the encoded sequence includes a transmembrane domain (amino acids 39 to 62). We cloned the NCU05136 gene, expressed a His6-tagged version of the protein in *N. crassa*, and purified the protein. We examined the enzymatic activity of the purified enzyme and were able to demonstrate with a TLC analysis and a PACE gel analysis that the enzyme cleaves a polysaccharide found in the growth medium from wild-type cells. We concluded that NCU05136 has enzymatic activity. The major digestion product identified in the TLC analysis and PACE gel analysis have an apparent size of an oligosaccharide with 4 or 5 sugar residues, indicating that the enzyme is an endo glycosylhydrolase. We found that the purified enzyme did not have activity on p-nitrophenyl-α-D-galactopyranoside, consistent with the enzyme functioning as an endo hydrolase. Without a highly purified substrate, we were unable to further characterize the enzyme, but our observations are consistent with the enzyme serving as an endo-α-1,4-galactosaminidase capable of digesting a GAG.

We showed that the deletion mutant for the deacetylase produced some extracellular polysaccharide that was soluble after ethanol precipitation containing galactosamine and/or *N*-acetylgalactosamine residues (Table 3). The soluble polysaccharides from the wild type lacked galactosamine and/or *N*-acetylgalactosamine residues. The results showed there are differences between the polysaccharides produced by the wild type and mutants. We suggest that the deacetylated GAG from the wild type may remain in the insoluble polysaccharide fraction, some of the acetylated GAG from the deacetylase mutant was found among the soluble polysaccharides, and that the deacetylase is active on the GAG.

We were able to obtain compositional information about the polysaccharides in the growth medium by concentrating the growth medium via evaporation and carrying out an analysis that discriminates between galactosamine and *N*-acetylgalactosamine. The analysis showed that the polysaccharides from the GAG synthase contained much less galactosamine and *N*-acetylgalactosamine than the polysaccharides from the wild type and are consistent with the synthase producing a GAG. The analysis demonstrated that *N. crassa* produces multiple polysaccharides containing galactose, galactosamine, and *N*-acetylgalactosamine sugars. With the medium containing multiple polysaccharides, we were unable to demonstrate compositional differences between the deacetylated GAG found the medium from the wild type and the acetylated GAG found in the medium from the deacetylase mutant.

The adhesion assays comparing the adhesive properties of deacetylase mutant hyphae with the acetylated GAG and the hyphae from the wild type were informative. We found that the wild-type hyphae provided adhesion to a glass surface, while the mutant hyphae were easily dislodged from the glass surface. This is similar to the findings of others, who showed that the deacetylated GAG provided adhesion to a variety of surfaces, including plastics and insect cuticle. We also found that wild-type hyphal mats had greater tensile strength than mutant hyphal mats, suggesting that the deacetylated GAG functions to give some cell-to-cell adhesion. Deacetylated GAGs were found by others to be important for cell-to-cell adhesion (clumping in liquid medium) and biofilm formation [5,7,9,12,14,15]. However, we found that the acetylated GAG produced by the deletion mutant was also an adhesive. In particulate binding assays, we found that the fully acetylated GAG functioned as an adhesive to alumina, celite (diatomaceous earth), activated charcoal, and wheat grass particulates. We hypothesize that the fully acetylated GAG is produced near or at the hyphal tips located near the growing edge of a hyphal colony, where it can bind to materials such as chitin, alumina, celite, activated charcoal, and wheat grass particulates. As the colony continues to grow, the GAG is deacetylated and its adhesive properties change. The GAG becomes capable of binding other materials such as other hyphae and glass. Our observations indicate that adhesive properties of the GAG are a function of the deacetylation process and of the location of the GAG relative to the growing edge of the *N. crassa* colony. This suggests a model for how GAG might be functioning within a growing fungal colony. We hypothesize that at the edge of the growing colony the newly synthesized acetylated GAG binds to newly synthesized chitin and helps anchor the GAG onto the cell wall instead of having the GAG being immediately released into the surrounding liquid. We further hypothesize that the acetylated GAG at the colony edge allows saprophytic and plant pathogenic fungi to bind to wheat grass and other similar food particulates they might find in their environment. This will facilitate the digestion of these particulates by having the particulates held tightly to the hyphae surface where the concentration of digestive hydrolases would be highest and by having their sugars and amino acids released near the plasma membrane to enhance the uptake of the nutrients. The binding of the growing edge of a hyphal colony to activated charcoal (burnt wood) by the acetylated GAG is particularly meaningful for *N. crassa*, which colonizes burnt trees after a forest fire. Adhering to the surface of a burnt tree would be helpful to *N. crassa* as it establishes itself in a natural environment. We hypothesize that when the acetylated GAG adheres to such particulates, it becomes protected from the deacetylase and the particulates remain adherent to the fungus. GAG not involved in such adhesions are then deacetylated to facilitate other interactions occurring farther back from the colony edge, including cell-to-cell interactions that increase the tensile strength of the hyphal network and the formation of biofilms. The GAG can be thought of as a “malleable glue” which allows the fungus to adhere to different types of surfaces and materials at different locations within a growing colony.

## Figures and Tables

**Figure 1 microorganisms-12-01509-f001:**
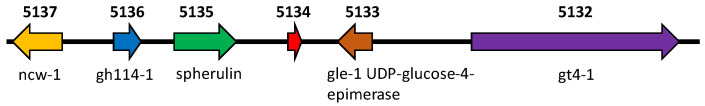
GAG gene cluster. The genome region of *N. crassa* containing the GAG biosynthetic genes from chromosome III is shown with the NCU numbers above the genes and the gene names below the genes.

**Figure 2 microorganisms-12-01509-f002:**
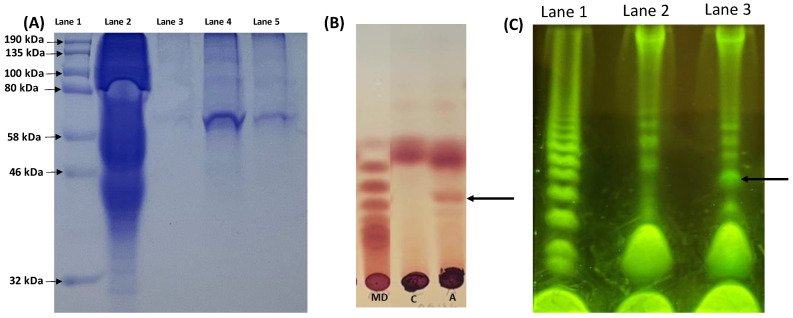
GAG hydrolase activity. (**A**) Purification profile of 5136::H6 using Ni-NTA column. Lane 1: Marker; Lane 2: whole protein extract; Lane 3: Elution fraction 7; Lane 4: Elution fraction 9; Lane 5: Elution fraction 11. (**B**) TLC assay of hydrolase activity of 5136::H6 on the ethanol precipitated polysaccharides from wildtype medium (MD: maltodextrin marker, C: no enzyme control, and A: enzyme reaction). (**C**) PACE gel hydrolase activity of 5136::H6 on ethanol precipitated polysaccharides from wildtype medium. Lane 1: maltodextrin ladder, Lane 2: Control without enzyme (5136::H6), Lane 3: Assay mixture with 5136::H6 enzyme. Arrows point to hydrolytic product.

**Figure 3 microorganisms-12-01509-f003:**
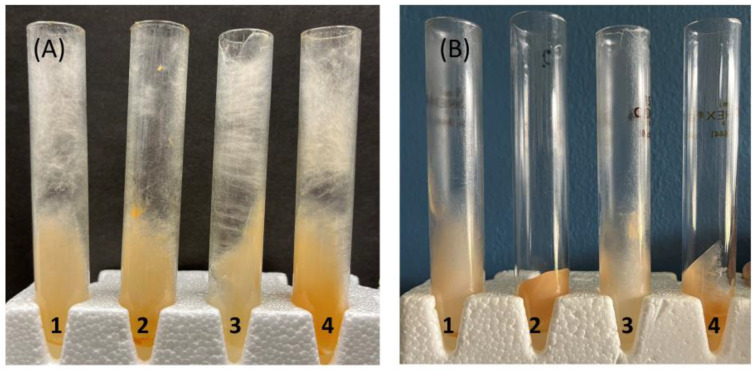
Glass binding assay: The ability of hyphae to remain adherent to glass during centrifugation was assessed. Tube 1 contains wildtype hyphae. Tube 2 contains ∆NCU05132 (GAG synthase) hyphae. Tube 3 contains ∆NCU05135 (GAG hydrolase) hyphae. Tube 4 contains ∆NCU05137 (ncw-1) hyphae. The hyphae were grown in centrifuge tubes for four days and subjected to centrifugation at 2500× *g*. (**A**) is an image of the tubes prior to centrifugation. (**B**) is an image of the tubes after centrifugation. Note that the hyphae from the GAG synthase mutant (tube 2) and the NCW-I GAG deacetylase mutant (tube 4) lost their adhesion to the glass during centrifugation.

**Figure 4 microorganisms-12-01509-f004:**
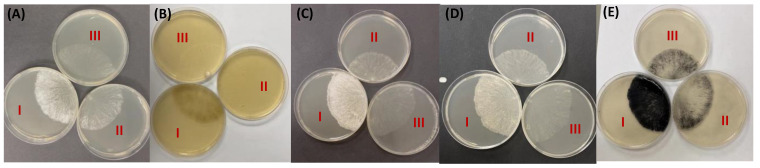
Particulate binding assay: (**A**) alumina (**B**) wheat grass (**C**) celite (**D**) chitin (**E**) activated charcoal. Petri dish I: ∆NCU05137 (deacetylase of Galactosaminogalactan mutant)) Petri dish II: wildtype, Petri dish III: ∆NCU05132 (galactosaminogalactan synthase mutant). ∆NCU05137 mutants bind to all the particulates whereas wildtype and ∆NCU05132 (GAG synthase mutant) have minimal binding.

**Table 1 microorganisms-12-01509-t001:** Strains used in experiments.

Strain	FGSC#
74-OR-S a (wild-type control strain)	4200
74-OR-23 A (wild-type control strain)	2489
∆NCU05132 a (∆GAG synthase)	14561
∆NCU05132 A (∆GAG synthase)	14562
∆NCU05137 a (∆GAG deacetylase)	11681
∆NCU05137 A (∆GAG deacetylase)	11682
∆NCU05135 a (∆GAG hydrolase)	15065
∆NCU05135 A (∆GAG hydrolase)	15066
∆NCU05136 a (heterokaryon) (∆GAG hydrolase)	13708

**Table 2 microorganisms-12-01509-t002:** Primers used for cloning and gene deletion analyses.

Primer Name	Sequence
5136F	ACAGGATCCCGACGAACAATGGCGGAC
5136H6R	CTATTGAATTCAGTGATGGTGGTGGTGGTGAGAGGTATTCGCAACCTTGGTCTC
5132XF	ACTTCACGCTGTGAGTGAGGC
5132XR	ACATGTGTGGATGAGATGTCG
5136XF	AAACCAGTGGTTGTGCCGTCC
5136XR	TACGGTATGCGCTATGTAGCC
5137XF	CTGTCTTGTAGGACTGAGTCG
5137XR	TGTACTGGTACGAGAGCTCGC
HygroR	CATATGCGCGATTGCTGATCC

**Table 3 microorganisms-12-01509-t003:** Compositional analysis of soluble polysaccharides after ethanol precipitation.

Sugar Linkage	∆NCU05132 GAG Synthase	Wild Type	∆NCU05137 GAG Deacetylase
Glucose	38.6%	6.6%	20.7%
Mannose	41.2%	67.5%	46.4%
Galactose	20.2%	25.9%	26.6%
*N*-acetylgalactose or galactosamine	<0.1%	<0.1%	6.4%

The composition of the polysaccharides from growth media that were soluble following an ethanol precipitation was determined. The mole percentage for each of the observed sugars is shown.

**Table 4 microorganisms-12-01509-t004:** Compositional analysis of polysaccharides from growth media.

Sugar Residue	∆NCU05132 GAG Synthase	Wild Type	∆NCU05137 GAG Deacetylase
Glucose	18.7%	8.5%	21.5%
N-acetylglucosamine	00.3%	00.5%	00.2%
Mannose	61.3%	68.4%	59.9%
Galactose	19.4%	17.7%	17.3%
Galactosamine	00.1%	02.6%	01.0%
*N*-acetylgalactosamine	00.1%	02.3%	00.1%

## Data Availability

Data is contained within the article or Appendix A.

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
