# Peer review of "Characterization of the Neurospora crassa Galactosaminogalactan Biosynthetic Pathway"

_microorganisms, 2024, doi:10.3390/microorganisms12081509_

Round 1

Reviewer 1 Report

Comments and Suggestions for Authors

In filamentous fungi, galactosaminogalactans (GAGs) have been shown to be crucial cell wall elements.

In most fungi, the enzymes for the GAG biosynthetic pathway are encoded in a five gene cluster. To characterize the cell wall of the filamentous fungus Neurospora crassa the authors asked what functions GAG might have in the cell wall and the culture medium of N. crassa.  The N. crassa GAG cluster encodes 5 enzymes for the GAG biosynthetic pathway and in addition one protein of unknown function: 1) GAG synthase for the synthesis of an acetylated GAG (NCU05132), 2) UDP-glucose-4-epemerase, (NCU05133), 3) protein of unknown function (NCU05134), 4), ) GH135-1, a GAG hydrolase with specificity for N-acetylgalactosamine-containing GAG (NCU05135), 5) GH114-1, a galactosaminidase with specificity for galactosamine-containing GAG (NCU05136), and 6) GAG deacetylase (NCW-1/NCU05137). Homokarytic deletion mutant isolates for NCU05132, NCU05135, NCU05136, and NCU05137 were ordered from the N. crassa deletion library and deletions were verified by PCR. Deletion mutant ΔNCU05136 turned out to be a heterokaryon and was not further characterized.

Analysis of polysaccharides found in the growth medium of the GAG synthase mutant (ΔNCU05132) demonstrated a major reduction in the levels of polysaccharides containing galactosamine and N-acetyl. This result provides evidence that the synthase is responsible for the production of a GAG.

Phenotypic characterization of wildtype and mutant isolates showed that deacetylated GAG from the wildtype can function as an adhesin to a glass surface and provides the fungal mat with tensile strength, demonstrating that the deacetylated GAG functions as an intercellular adhesive. The acetylated GAG produced in increasing amounts by the deacetylase mutant (ΔNCU05137) was found to function as an adhesive for chitin, alumina, diatomaceous earth, activated charcoal, and wheat leaf particulates.

The enzymatic activity of the GAG hydrolase encoded by NCU05136 demonstrated after purification of a HIS-tagged version of the enzyme from N. crassa.

The study is interesting and of importance in the field of fungal cell wall biology and function. The data presented in the manuscript mostly are solid and convincing.

However, a major weakness of the analysis is that no complemented strains where used as control. In addition, the nomenclature of the mutants used should be more uniform so as not to confuse the reader. It is not made clear why some analyses were only done with some of the mutants.

- 10- 12: it would be better to name all five enzymes include also NCU05133.

- 11: (NCW1/NCU05137) instead of (/NCW-1/NCU05137)

- Table 1 and 90-97: it is incomprehensible that the knockout of NCU05135 was not verified by PCR. PCR verification of knockouts should be shown in the supplement.

- In the section Materials  a table of strains used in this study with their characteristics should be included in the manuscript.

- Test for tensile strength of a hyphal mats should be described in more detail. What is order number of the penetrometer, which tip was used, how was the home-made plastic apparatus used for measuring tensile strength constructed. Fig. S1 should show the hyphal mat in the penetrometer.

- 168: The abbreviation of TLC and PACE should be explained when they are used for the first time

- 269-271: not all mutants were analyzed only NCU05132, NCU05135 and NCU05137.

- 273-275: it should be described which mutants were analyzed by PCR

- Fig. 2  B:  seems to be a composite image. Marker and enzyme assays do not appear to come from the same gel. C: it is not clear why the precipitated oligosaccharides are from ΔNCU05135.

Fig. 3.and Fig. 4: Complemented strains of the mutants should be  shown as control.

The discussion contains some repetitions that should be avoided e.g. see 419-420 and 437-439.

Author Response

We have made several changes to our manuscript “Characterization of the Neurospora crassa galactosaminogalactan biosynthetic pathway” in response to suggestions from the reviewers.   Below is a detailed accounting of these changes.   The changes are found in red type on the revised manuscript so that they can be easily identified.

We would like to thank the reviewers for their efforts and feel that their comments have helped improve the manuscript.   Sincerely yours, Stephen J. Free, corresponding author

Reviewer 1.

  1. Reviewer 1 indicated that he/she felt a major weakness was that we did not include complemented strains as controls. This would be very difficult to do for the GAG synthase gene, which is 9,500 bases in length.  These genes have been characterized in Aspergillus fumigatus and Aspergillus oryzae, and it is clear from our data that we have the deletion mutants.  Thus, we feel that it is appropriate to use the wildtype strain for a control in our experiments. 
  2. Reviewer 1 wanted the nomenclature to be more uniform. In the manuscript we have use the NCU#’s and we’re not sure what the reviewer is alluding to by this comment, but we have added the NCU# to the manuscript in places where there could be a confusion.
  3. Reviewer 1 commented that it wasn’t clear why some analyses were only done with some of the mutants. The two mutants which gave clear phenotypic results were the GAG synthase mutant (NCU05132) and the GAG deacetylase mutant (NCU05137).   Since the other mutants didn’t have clear phenotypes, we did not include them in the report. The one exception to this is the glass binding assay (centrifugation assay), which did include the NCU05135 mutant.  The manuscript does include a GAG hydrolase assay for NCU05136, but we do not have a mutant for this gene.
  4. In lines 10-12 the reviewer asked that we include all five enzymes in the cluster, and we added NCU05133 per the reviewer’s suggestion.
  5. In line 11 the reviewer suggested that we use NCW1 instead of NCW-1. The NCW-1 is correct nomenclature for crassa proteins, and so we haven’t made the change.
  6. Reviewer 1 commented that we didn’t include a PCR analysis demonstrating that the NCU05135 strain had KO deletion. Since NCU05135 is not being phenotypically characterized we didn’t include a KO analysis for it.  
  7. In response to the request from Reviewer 1, we have added Figure S1 to the manuscript showing the PCR analysis for the two deletion mutants being phenotypically characterized.
  8. Reviewer 1 asks that a Table of strains be included in the Materials and Methods. We have added Table 1 with a listing of the strains to the Materials and Methods.   
  9. Reviewer 1 asked that we provide more information about how the tensile strength of the hyphal mats was tested. We have added a couple of sentences to provide this information.  The manuscript now reads “The penetrometer (model FT 02) was purchased from QA Supplies (Norfolk, VA).  The home-made apparatus was constructed from 2 sheets of plexiglass with an array of nine 3 mm holes drilled into the plexiglass sheets.  A 2 mm probe was used in the penetrometer for measuring hyphal tensile strength.”  We have added a new Figure of the penetrometer with a hyphal mat inside as requested by Reviewer 1.
  10. Reviewer 1 asked that we explain the abbreviations TLC and PACE where they were used for the first time. We have done so, and the text now reads “Assays for GH114-1 hydrolase activity by TLC (thin layer chromatography) and PACE (polysaccharide analysis using carbohydrate gel electrophoresis)”.
  11. Reviewer 1 indicates that not all mutants were analyzed in the PCR assay for KO deletions. We have rewritten this section to clarify that only NCU05132 and NCU05137 were analyzed. 
  12. Reviewer 1 indicated that Figure 2B seems to be a composite figure. The Figure is from a TLC plate and the lanes were all from the same plate/experiment.  However, on the original TLC plate there were a few lanes between the maltodextrin marker and the experimental samples.  In preparing the image for publication, we “deleted” the lanes between the marker and the experimental samples being shown. 
  13. Reviewer 1 commented on why the precipitated oligosaccharides are from NCU05135 – we assume the reviewer was referring to Figure 2C where we had written the oligosaccharides were from NCU05132. We had made an error in preparing the legend for the figure and the oligosaccharides were from the wildtype.  We have changed this information in the figure legend.
  14. In Figures 3 and 4 Reviewer 1 asked that we use complemented strains. As indicated above, this would be very difficult to for the GAG synthase mutant, and we feel that the wildtype strain is an appropriate control. 
  15. Reviewer 1 noted that there is some repetition in the discussion section with information on lines 419-429 being repeated in lines 437-439. We have deleted the sentence from lines 437-439 in response to this suggestion.  

Reviewer 2 Report

Comments and Suggestions for Authors

This paper describes the characterization of the Neurospora crassa galactosaminogalactan biosynthetic pathway. It provided evidence that the GAG synthase is responsible for the production of a GAG. It also indicated that there are other galactose-containing polysaccharides. The phenotypic analysis showed that deacetylated GAG from the wildtype can function as an adhesin to a glass surface and provides the fungal mat with tensile strength. The acetylated GAG produced by the deacetylase mutant was found to function as an adhesive. The results have significant implications. some points needed revised are mentioned below.

p.1, line 9: “The Neurospora genome”… should be “”Neurospora crassa genome…. 

The letter N for the word “N-acetylgalactosamine” in the text should be italic.

p.9, Figure 2A: The molecular weight for each marker in line 1 should be given. 

p.12, Line 446: “is was…” should be “it was…”. 

Author Response

We have made several changes to our manuscript “Characterization of the Neurospora crassa galactosaminogalactan biosynthetic pathway” in response to suggestions from the reviewers.   Below is a detailed accounting of these changes.   The changes are found in red type on the revised manuscript so that they can be easily identified.

We would like to thank the reviewers for their efforts and feel that their comments have helped improve the manuscript.   Sincerely yours, Stephen J. Free, corresponding author

Reviewer 2

  1. Reviewer 2 asked that we change “Neurospora genome” to “Neurospora crassa genome”. We have made the suggested change.
  2. Reviewer 2 indicated that the N in N-acetylgalactosamine should be in italics. We have made this change throughout the manuscript.
  3. Reviewer 2 asked that we include the molecular weights for each of the markers in Figure 2A. We have added the molecular weights as requested.
  4. Reviewer 2 noted a typographical error (“is was” should be “it was”). We have corrected the error.

Reviewer 3 Report

Comments and Suggestions for Authors

The biosynthesis of the fungal polysaccharide galactosaminogalactan (GAG) has been studied in some detail for Aspergillus fumigatus but not so much for plant fungi such as Neurospora crassus. This paper describes a preliminary characterisation of the N. crassus gene cluster containing the biosynthetic enzymes for this polysaccharide, with a particular focus on the properties of deletion mutants lacking either the GAG synthase or the GAG deacetylase. A galactosaminidase gene product was expressed and found to have enzymic activity on polysaccharides isolated from the growth medium.

Polysaccharides were prepared from the growth medium of wt and mutants, but among these the proportion of GAG was small compared with mannose and glucose based polysaccharides, and galactose polysaccharides other than GAG. However it was clear that the synthase is essential for GAG biosynthesis. The function of the deacetylase is less clearcut, though the deletion mutant may produce a soluble N-acetylated polysaccharide.

The adhesion behaviour of the mutants is interesting. Hyphae of the wt and mutants have differing adhesion profiles. Wt, but not either of the synthase or deacetylase mutants, bind to a glass surface., and wt hyphal mats had higher tensile strength than those of the mutants. In a simple petri dish assay, deacetylase mutant hyphae appear to bind more strongly to a range of materials than wt or synthase mutant.

This study was clearly of a preliminary nature, and not all of the results are neat and conclusive. It would obviously be better to isolate pure GAG and characterise its structure in more detail. With some minor changes though the work may be publishable and may provide a basis for further work in the future.

Comments:

1.       Quantitative data should be expressed as mean +/- SD or SE, and the number of independent estimations n should be given. This applies to Tables 2 and 3, and the data in the text on lines 360-362.

2.       Non-quantitative data: with respect to the results presented in Fig. 4, presumably this experiment has been performed several times and the results shown are a typical example? Please give details. It is particularly important that these results are shown to be repeatable, as the speculative discussion relies on them quite heavily.

Author Response

We have made several changes to our manuscript “Characterization of the Neurospora crassa galactosaminogalactan biosynthetic pathway” in response to suggestions from the reviewers.   Below is a detailed accounting of these changes.   The changes are found in red type on the revised manuscript so that they can be easily identified.

We would like to thank the reviewers for their efforts and feel that their comments have helped improve the manuscript.   Sincerely yours, Stephen J. Free, corresponding author

Reviewer 3

  1. Reviewer 3 asked that we indicate that the quantitative data be expressed as mean +/- SD and that we indicate the number of independent estimations. We have added this information to the text.
  2. Reviewer 3 asked that we apply the mean +/- SD information for Tables 2 and 3 (now Tables 3 and 4). The information in these tables comes from a carbohydrate analysis carried out at the CCRC and the information was supplied as the results of a single analysis, so we don’t have mean +/- SD information for the information in these tables.
  3. Regarding the non-qualitative data in Figure 4, Reviewer 3 asked that we provide details about how many times the experiments were done. We have added this information to the text for the penetrometer data and for Figures 3 and 4.  For the penetrometer assay we added “Four different series of these penetrometer assays were carried out with similar results to demonstrate that the results were reproducible”.  For the centrifugation assay (Figure 3) we added “This centrifugation assay to assess the ability of the hyphae to retain adherence to the centrifuge tube was repeated six times to demonstrate the results shown in Figure 3 were reproducible”.   For the particulate binding assay (Figure 4), we added “Five independent particulate binding experiments were carried out to demonstrate that the binding of these particulates was reproducible”. 

Round 2

Reviewer 1 Report

Comments and Suggestions for Authors

Although complementation of knock-out strains would be the better control, I understand, that the construction of these strains is very difficult.